# Enzymatic hydrolysis of starch from the anthocyanin extraction residue (AER-starch) with ultrasound pretreatment: A techno-economic assessment

J.D. Quiñonez-Ensuncho, Susana Ochoa[☯], Víctor-Manuel Osorio-Echeverri[iD][☯],
J. Felipe Osorio-Tobón[iD]*

Faculty of Health Sciences, University Institution Colegio Mayor de Antioquia, Medellín, Antioquia, Colombia

☯ These authors contributed equally to this work.
* juan.tobon@colmayor.edu.co

## Abstract

Purple yam (*Dioscorea alata*) is a tropical tuber crop that is a source of anthocyanins and starch. Starch can be isolated from the large quantity of extraction residues and become a raw material for hydrolyzate production. Ultrasound pretreatment was evaluated in consecutive enzymatic hydrolysis (UCEH) and simultaneous enzymatic hydrolysis (USEH) of starch from anthocyanin extraction residue (AER-starch). A techno-economic assessment was performed. Ultrasound pretreatment showed a positive influence on CEH and SEH. The hydrolysis degree (HD) was higher in CEH (72.87 ± 3.01%) than in SEH (28.10 ± 1.32%). CEH performed better than SEH, primarily due to the thermal instability of the enzymes. Enzymatic hydrolysis solubilized almost all AER-starch. Starch conversion (SC) above 90% was achieved for all CEH experiments using AER-starch concentrations ranging from 10 to 50 g L$^{-1}$. The cost of manufacturing (COM) for glucose syrup using AER concentrations of 10 and 350 g L$^{-1}$ was US\$61.25 kg$^{-1}$ and US\$2.90 kg$^{-1}$, respectively, when the purchasing cost for AER-starch was US\$1.50 kg$^{-1}$. The CRM was the major component of the COM (above 50%) when AER-starch concentrations were higher than 100 g L$^{-1}$, AER-starch being the primary raw material contributing to the COM. The production of the glucose syrup was economically feasible when the AER-starch purchasing price was under US\$1.20 kg$^{-1}$ and the selling price was at least US\$3.50 kg$^{-1}$.

## Introduction

Yam (*Dioscorea* spp.) is a tuber plant consumed in tropical regions of South America, Asia, Africa, and the Caribbean [1]. Purple yam (*Dioscorea alata*) is a type of yam that is more cultivated due to its characteristics. Besides the average composition of yam, which contains starch (16–24%), proteins (1.3–6.9%), cellulose (1–7.5%), sugar

**Data availability statement:** The complete dataset is openly accessible on Zenodo at the following DOI: https://doi.org/10.5281/zenodo.18257051.

**Funding:** Institución Universitaria Colegio Mayor de Antioquía. Convocatoria: 2019 - CONVOCATORIA INSTITUCIONAL PARA LA CONFORMACIÓN DEL BANCO DE PROYECTOS DE INVESTIGACIÓN CIENTÍFICA, DESARROLLO TECNOLÓGICO E INNOVACIÓN.

**Competing interests:** The authors have declared that no competing interests exist.

(0.5–1.2%), fats (0.05–0.2%) and water (65–78%), it contains a group of antioxidant compounds known as anthocyanins responsible by its red-to-purple color [2,3]. Purple yam could be a valuable source of anthocyanins and starch. For example, as anthocyanins are water-soluble compounds, they can be used as natural colorants. However, anthocyanin extraction generates large amounts of residues, where almost all the starch is retained, making this waste an excellent source of this valuable compound. Thus, starch isolation allows the obtaining of an ingredient widely used in the industry. Moreover, the starch obtained from the anthocyanin extraction residue (AER-starch) could be a valuable source of starch hydrolysates that could facilitate the valorization of purple yam as a crop and contribute to the circular economy due to the intensive use of this raw material.

Starch hydrolysates can be used primarily as substrates in various fermentation processes related to the food and beverage industries. The enzymatic transformation can involve the use of α-amylase and amyloglucosidase. Generally, this transformation is performed in two steps: liquefaction using α-amylase, followed by saccharification catalyzed by glucoamylase. Industrially, the process is carried out in two stages: first, the starch is gelatinized at 90°C, and then the temperature is reduced to approximately 60°C for saccharification [4]. Although enzymatic hydrolysis is well-established, further research is needed to enhance hydrolysis yields. A common approach is through the application of a pretreatment. Ultrasound-assisted pretreatment enables the enzyme to access the starch molecule, thereby increasing hydrolysis yields. An acoustic phenomenon known as cavitation is caused by ultrasonic waves, producing bubbles that release a high amount of energy when they implode. This energy disturbs the native starch granules, increasing their surface area, which enhances enzyme accessibility and improves hydrolysis efficiency [5]. Ultrasound pretreatment has been successfully used in the enzymatic hydrolysis of starch [6,7] as well as in other substrates such as microalgae [8], brewer´s spent grains [9], and rice [10], among others.

Depending on the matrix and enzyme, the ultrasound operation parameters have different effects. For example, during the preparation of hydrolyzed egg yolk powder after ultrasound pretreatment, cavitation and excessive ultrasound led to gradual increases and decreases in surface hydrophobicity and free sulfhydryl groups, as observed with increasing ultrasound time. In consequence, the protein swollen with disulfide bonds becomes more aggregated, and the reconstruction of these bonds was facilitated by cavitation [11]. Cavitation also generates microjets and radicals, thereby increasing selective disruption of the cell wall in lignocellulosic matrices such as brewer's spent grains. Moreover, certain combinations of ultrasound power and frequency can induce structural reorganization, which restricts cellulose accessibility [9]. For example, in that study, the highest cellulose concentration was observed at 130 kHz and 200 W, whereas the lowest was observed at 5 kHz and 100 W. Another factor with a key role in the cavitation phenomenon is the ultrasound intensity, which influences bubble nucleation, growth, and collapse. Ultrasound intensities ranging from 0.5 to 8 W mL$^{-1}$ were evaluated in the pretreatment enzymatic hydrolysis of polygalacturonic acid [12]. Excessively high intensities increase fluid circulation, leading to reduced cavitation efficiency due to foam formation and excessive air entrainment, particularly at 8 W mL$^{-1}$.

On the other hand, economic feasibility analysis helps identify bottleneck operations within the processes and high-lights critical financial factors for reducing manufacturing costs. Economic evaluation of the ultrasound-assisted pretreatment and enzymatic hydrolysis of the starch obtained from the extraction residue is a key factor in scaling up the process. SuperPro Designer® (Intelligen Inc., Scotch Plains, NJ, USA) is a flowsheet-driven simulator used for modeling processes in the food and biotechnology industries, among others. The software incorporates the composition and purchase costs of raw materials, utilities, waste treatment, labor, and royalties to perform cost analyses and economic evaluations. It also facilitates material and energy balances, equipment sizing, and environmental impact assessments [13].

This work corresponds to the third step of a project aimed at promoting the purple yam crop in Colombia. In the first step, the anthocyanins were extracted [14]. In the second step, the starch was isolated from the extraction residue [15]. In the third step (this work), ultrasound as a pretreatment was used to enhance the enzymatic hydrolysis of the starch. Therefore, the objective of this work was to evaluate the enzymatic hydrolysis of starch isolated from the anthocyanin extraction residue using ultrasound-assisted pretreatment, along with a techno-economic analysis.

## Materials and methods

### Materials

The AER-starch was obtained from the purple yam anthocyanin extraction residue, following the methodology described in previous work [15]. Industrial α-amylase from *Bacillus licheniformis* (120,000 U g$^{-1}$) (HS-120, BIOKATAL) and amylo-glucosidase from *Aspergillus niger* (130,000 U g$^{-1}$) (H – NHY Enzymes & S N Chemical Co. Ltd) were purchased from Proenzimas (Calí, Colombia). All chemicals used were analytical grade.

### Consecutive enzymatic hydrolysis (CEH) and simultaneous enzymatic hydrolysis (SEH)

Consecutive enzymatic hydrolysis (CEH) was performed according to the methodology described by Almeida et al. [16], with slight modifications. An AER-starch 16 mM sodium acetate buffer solution with a concentration of 10 g L$^{-1}$ was hydrolyzed under the conditions recommended by the enzyme supplier. In the liquefaction stage, α-amylase was added at a dosage of 180 U g$^{-1}$ of AER-starch, and the mixture was incubated at 90°C for 120 minutes. Subsequently, the temperature and pH were adjusted to 60°C and 4.5, respectively, and amyloglucosidase (120 U g$^{-1}$ of AER-starch) was added. The reaction was then incubated for an additional 120 minutes to facilitate saccharification. Enzymatic activity was stopped by boiling the mixture, followed by centrifugation at 3620 × g for 30 minutes (Hermle, Z326K, Germany). The supernatants were stored at –4 °C for further analysis, and the precipitate was dried in an oven at 60 °C for 24 hours (Memmert, model UN110, Schwabach, Germany) to determine starch conversion (SC) using the following equation:

$$SC\ (\%) = \left( \frac{Initial\ starch\ (g) - dried\ precipitated\ (g)}{Initial\ starch\ (g)} \right) \times 100$$

(1)

In simultaneous enzymatic hydrolysis (SEH), α-amylase (180 U per gram of AER-starch) and amyloglucosidase (120 U per gram of AER-starch) were incubated simultaneously in 50 mL of 16 mM sodium acetate buffer (pH 5.0) at 75 °C for 120 minutes. The supernatants were obtained as previously described.

### Ultrasound-assisted pretreatment

Ultrasound was used as a pretreatment in the consecutive enzymatic hydrolysis process (UCEH) and the simultaneous enzymatic hydrolysis process (USEH). The AER-starch mixture was prepared according to the previously described methodology. Ultrasound pretreatment was performed using a 750 W ultrasonic homogenizer (Cole-Parmer, Vernon Hills, IL, USA) equipped with temperature control (30°C). AER-starch mixtures were treated for 30 minutes at an amplitude of

60% in pulse mode (2 seconds ON, 2 seconds OFF), with an estimated delivered power of 600 W and an acoustic energy density of 592 W cm$^{-2}$.

## Hydrolysis degree (HD)

3, 5-Dinitrosalicylic acid (DNS) was used to determine the reducing sugar yield, following the methodology described by Miller [17] with some modifications. Absorbance was detected at 540 nm using a spectrophotometer (Thermo Fisher Scientific, Madison, USA). The analyses were performed in triplicate. Equation (2) was used to calculate the hydrolysis degree (HD):

$$HD\ (\%) = \frac{Reducing\ sugar\ expressed\ glucose\ (g)}{Initial\ starch\ (g)} \times 100 \tag{2}$$

## Process simulation model

The process simulation and the techno-economic evaluation were performed using SuperPro Designer v14® (Intelligen Inc., Scotch Plains, NJ, USA) software. The process was modeled using the ideal thermodynamic property method, which assumes ideal mixing behavior. Physical properties of compounds such as water, starch, and glucose were obtained from the SuperPro Designer database. New components, such as the hydrolyzed starch fractions, were entered into the software using physical properties reported in the literature. The flowsheets for the enzymatic hydrolysis process and the enzymatic hydrolysis with ultrasound pretreatment process are shown in Fig 1 and Fig 2, respectively. The process consisted of two sections: enzymatic hydrolysis and purification. The AER-starch ultrasound-assisted pretreatment used an ultrasound unit (P-21/V104), a cluster of four 16 kW ultrasonic processors with temperature control. In the enzymatic

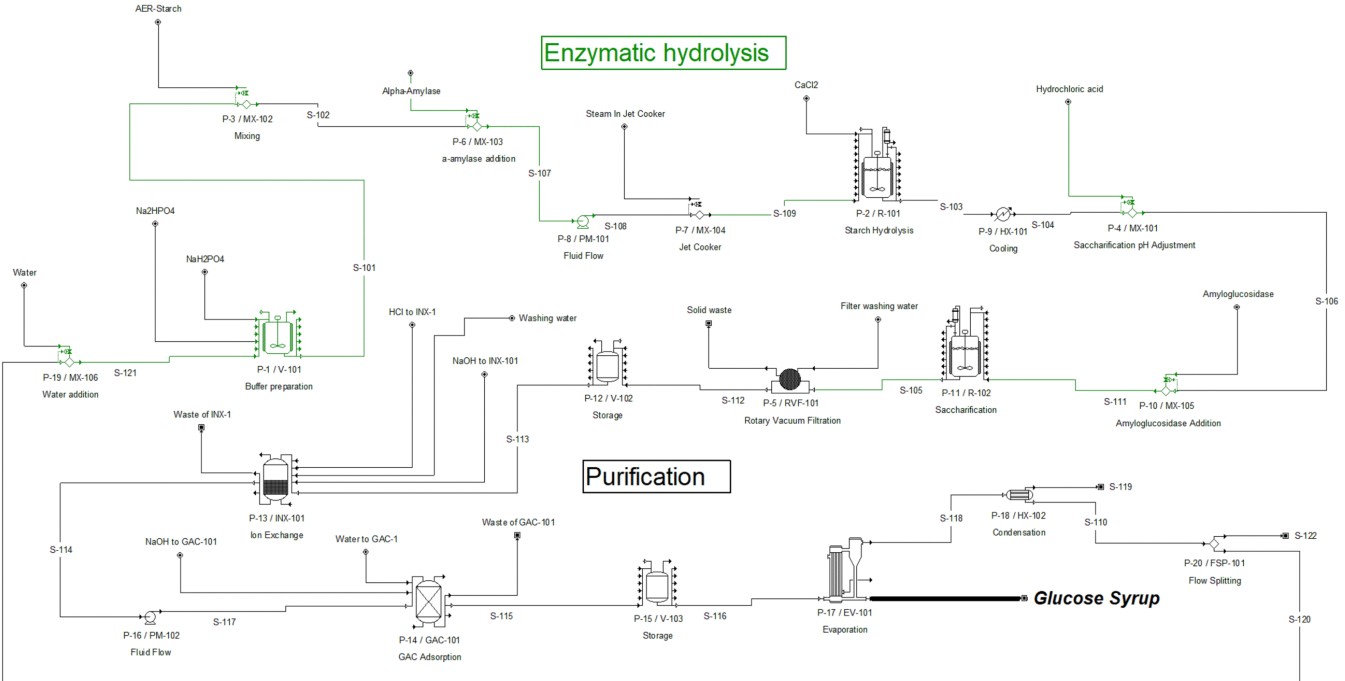

**Fig 1. Flowsheet of the enzymatic hydrolysis process using SuperPro Designer v14® software.**

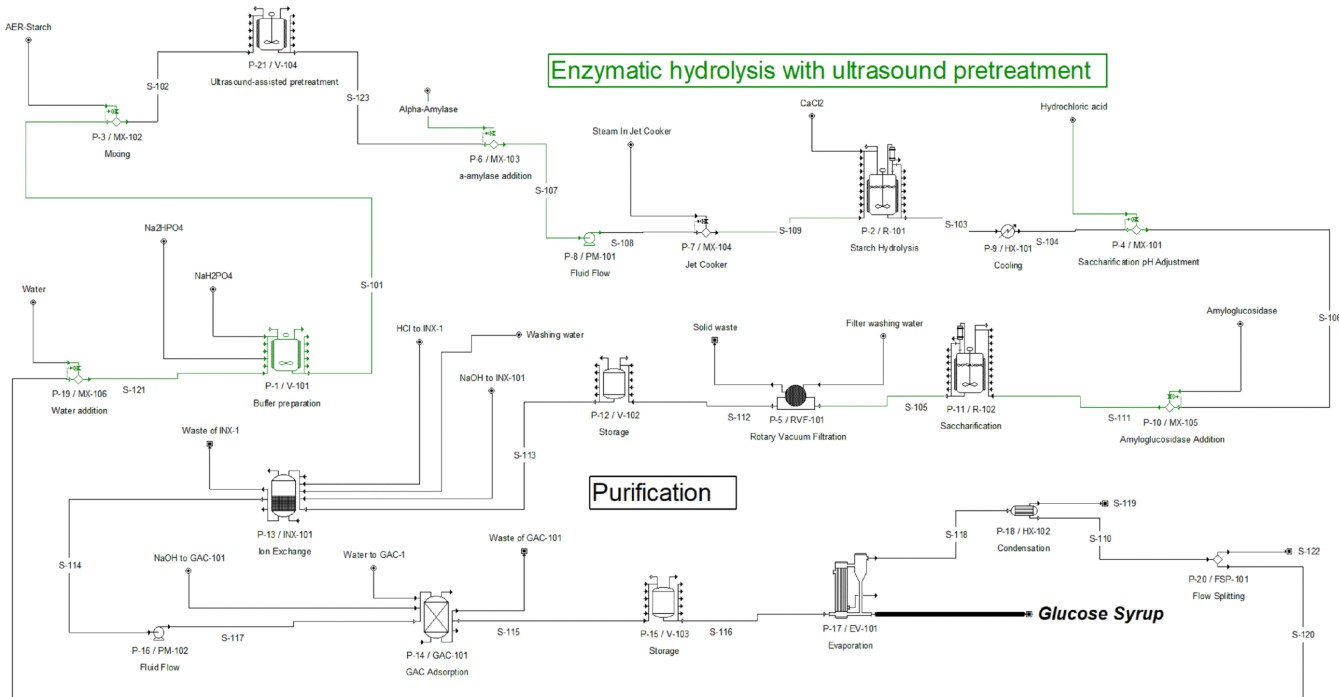

**Fig 2. Flowsheet of the enzymatic hydrolysis process with ultrasound pretreatment using SuperPro Designer v14® software.**

hydrolysis, the AER starch is mixed with the buffer (S-101) to achieve the desired concentration. The buffer is prepared in (V-101) using water, sodium phosphate dibasic ($Na_2HPO_4$), and sodium phosphate monobasic ($NaH_2PO_4$). Then, alpha-amylase is added (MX-103), and the mixture is heated at 90°C (MX-104). Next, the starch liquefaction is carried out in R-101. After liquefaction, the stream is cooled to 60°C (HX-101), and the pH is adjusted using hydrochloric acid in MX-119. Amyloglucosidase is then added (MX-105) for saccharification in the saccharification tanks (R102). For purification, salts, proteins, enzymes, and other compounds are removed. A rotary vacuum drum filter (RVF-101) removed proteins and other solids. Then, the syrup is stored in V-102, where the pH is adjusted to remove all salts in the ion exchanger (INX-101). Then, a buffer tank (V-103) stores the syrup before removing color, odors, and protein traces in the carbon column (GAC-101). Finally, the syrup is concentrated in a multi-effect evaporator (EV-101), obtaining glucose syrup (90% glucose).

Factors such as mixing and heat transfer, as well as viscosity and enzyme kinetics, should be considered during scale-up. For example, an increase in viscosity can reduce bubble collapse, reducing the cavitation effect and mass transfer rates. Moreover, the distribution of ultrasound energy is influenced by reactor size because cavitation is not uniform across the reactor. Moreover, an increase in temperature requires precise temperature control to preserve enzyme activity [18]. On the other hand, as acoustic cavitation may negatively affect enzyme stability, the ultrasound is commonly applied as a pretreatment rather than simultaneously.

In practice, perfect mixing conditions cannot be achieved at a large scale, and there is no single consensus approach for scale-up [19]. Instead, a scale-up approach could rely on maintaining comparable laboratory-scale operating conditions and assuming they can be extrapolated, while acknowledging unavoidable gradients in temperature, pH, and concentration at the industrial scale. In this approach, we assumed that the HD and SC of the hydrolysates obtained at the laboratory scale would be reproduced at larger scales under the same hydrolysis conditions (e.g., AER-starch concentration, hydrolysis time, temperature, etc.).

## Techno-economic evaluation

The system was designed to process 5,000 L per batch of AER-starch solution, operating 330 days per year. The AER-starch amount refers to the amount of starch added to reach the target concentration required for enzymatic hydrolysis. The purchase cost of the equipment was estimated using vendor quotations, previous studies, and data from the Super-Pro Designer database, along with a power-law equation cost model [20]. Table 1 summarizes the equipment nomenclature and description of equipment used in the enzymatic hydrolysis process. The cost of manufacture (COM) was determined by the sum of direct costs, fixed costs, and general expenses, based on fixed capital investment (FCI), cost of raw material (CRM), cost of operational labor (COL), cost of utilities (CUT), various consumables cost (VCT), and cost of waste treatment (CWT) [21]. FCI represents the equipment cost, process piping, instrumentation, and construction, among others. CRM includes the cost of all materials for AER-starch hydrolysis and purification. COL represents the cost of the operators, and CUT is the sum of the costs of power and heat transfer agents. VCT refers to the total cost associated with materials that are not reusable and must be replaced periodically (e.g., filters, resins, and other operational supplies). CWT represents the cost of treatment or disposal of liquid or solid wastes. The economic data employed to calculate the COM is presented in Table 2. The economic analysis assumptions for the enzymatic hydrolysis process are presented in S1 Table. In addition, S2 Table summarizes the mass and energy balances for the process.

## Sensitivity study

The feasibility of the process was assessed using a sensitivity analysis based on the following economic parameters: return on investment (ROI), gross margin, internal rate of return (IRR), net present value (NPV) at 7.00%, and payback time. Various scenarios were tested to evaluate the effect of the component (FCI, CRM, COL, CUT, CVT, or CWT) that has the highest influence on the manufacturing cost and the selling price of the glucose syrup on the project feasibility.

## Statistical analysis

Statistical analyses were performed using Minitab version 19, with a significance level of 0.05. Treatment means were compared using Tukey's test at a 5% significance level. All experiments were performed in duplicate.

**Table 1. Nomenclature and description of equipment used in the enzymatic hydrolysis process.**

| Quantity | Name | Description | Total cost (US$) |
|---|---|---|---|
| 1 | R101 | Stirred Reactor. Vessel Volume = 9277.88 L | 37,000.00 |
| 1 | INX-101 | Ion Exchanger. Column Volume = 1898.31 L | 24,000.00 |
| 1 | R-102 | Stirred Reactor. Vessel Volume = 8911.23 L | 36,000.00 |
| 1 | PM-102 | Centrifugal Pump. Pump Power = 0,39 kW | 18,000.00 |
| 1 | RVF-101 | Rotary Vacuum Filter. Filter area: 32.08 $m^2$ | 31,000.00 |
| 1 | V-101 | Blending Tank. Vessel Volume = 5295,13 L | 15,000.00 |
| 1 | EV-101 | Multi-Effect Evaporator. Mean Heat Transfer Area = 2.41 $m^2$ | 24,000.00 |
| 1 | V-102 | Receiver Tank. Vessel Volume = 8910.84 L | 15,000.00 |
| 1 | GAC-101 | GAC Adsorber. Column Volume = 287.90 L | 9,000.00 |
| 1 | V-103 | Receiver Tank. Vessel Volume = 10958.61 L | 16,000.00 |
| 1 | HX-102 | Condenser. Condensation Area = 258.27 $m^2$ | 13,000.00 |
| 1 | PM-101 | Centrifugal Pump. Pump Power = 0.3 kW | 4,000.00 |
| 1 | HX-101 | Heat Exchanger. Heat Exchange Area = 0.72 $m^2$ | 3,000.00 |
| – | – | Piping and connections | 61,000.00 |
| 1 | V-104 | UAE equipment. Energy = 16 kW. Processing capacity of 12 $m^3$ $h^{-1}$ | 339,284.00 |

**Table 2. Economic parameters used in the process simulation with SuperPro Designer 14.**

| *Fixed capital investment (FCI)* | |
|---|---|
| Equipment purchases cost [a] | US$301,000.00 |
| Installation [a] | US$121,000.00 |
| Instrumentation and buildings [a] | US$567,000.00 |
| *Cost of operational labor (COL)* | |
| Wage [b] | US$1.96 h$^{-1}$ |
| *Cost of raw material (CRM)* | |
| AER-Starch [c] | US$1.50 kg$^{-1}$ |
| Alpha-amylase [d] | US$41.39 kg$^{-1}$ |
| Amyloglucosidase [d] | US$70.60 kg$^{-1}$ |
| Calcium chloride ($CaCl_2$) [d] | US$7.06 kg$^{-1}$ |
| Disodium phosphate ($Na_2HPO_4$) [d] | US$9.66 kg$^{-1}$ |
| Monosodium phosphate ($NaH_2PO_4$) [d] | US$8.34 kg$^{-1}$ |
| Hydrochloric acid (HCl) [d] | US$1.87 kg$^{-1}$ |
| Sodium hydroxide (NaOH) [d] | US$2.54 kg$^{-1}$ |
| *Cost of utilities (CUT)* | |
| Electricity [e] | US$0.24 kWh$^{-1}$ |
| Water steam (high pressure) [a] | US$32.00 ton$^{-1}$ |
| Water [e] | US$1.53 m$^{-3}$ |
| *Various consumables cost (VCC)* | |
| Ion exchange resin [d] | US$2.00 kg$^{-1}$ |
| GAC Packing [d] | US$4.00 kg$^{-1}$ |
| *Cost of waste treatment (CWT)* | |
| Aqueous waste [e] | US$1.17 m$^{-3}$ |
| Solid waste [e] | US$91.03 ton$^{-1}$ |

[a] SuperPro Designer 14® Database.

[b] Ministerio del trabajo de Colombia (Bogotá, Colombia).

[c] Éxito. Cassava starch. Retrieved from: https://www.exito.com/yucarina-49999/p.

[d] Direct quotation.

[e] Empresas públicas de Medellín (Medellín, Colombia).

## Results and discussion

### Effect of ultrasound pretreatment on CEH and SEH

Table 3 shows the HD and SC of hydrolysates obtained from AER-starch. In this work, the HD ranged between 11.43 ± 1.11% and 72.87 ± 3.01%. The literature has reported similar or lower values of HD for other sources of starch. For example, these results are superior to those found in the ultrasound-assisted enzymatic processing of yam using amylases [22]. In that work, the HD ranged between 9.70% and 20.50% after 240 minutes of enzymatic hydrolysis. However, as the enzymatic hydrolysis was carried out in fresh yam, the availability of starch is likely lower than in isolated starch, such as AER-starch. Additionally, the fresh yam has a lower starch concentration, as well as a higher amount of other compounds, such as fiber and proteins, which can interfere with the action of the enzymes. In another study, Jin et al. [23] achieved an HD of around 20% in the enzymatic hydrolysis of potato starch using amyloglucosidase at 75°C for 50 min. In this work, as the processing time was longer, the enzymes could hydrolyze more starch molecules, obtaining a higher HD.

According to the literature, the use of ultrasound as pretreatment enhances the performance of the hydrolysis process [9,24]. The results obtained in this work indicate that the conversion of AER-starch into fermentable sugars was

**Table 3. Summary of SC and HD from CEH, SEH, UCEH, and USEH.**

|  | HD (%) | SC (%) |
|---|---|---|
| CEH | 67.93±6.31[A] | 92.66±2.46[A] |
| SEH | 11.43±1.10[C] | 54.75±2.66[B] |
| UCEH | 72.87±3.01[A] | 96.77±0.15[A] |
| USEH | 28.10±1.32[B] | 91.25±0.55[A] |

Means within columns followed by different letters are significantly different (p<0.05).

almost complete, mainly when ultrasound was used as a pretreatment in the CEH. The acoustic phenomenon of cavitation causes the structural breakdown, and a larger quantity of starch polymers is solubilized, improving the efficiency of the enzymatic reaction. The application of ultrasound as a pretreatment increased the HD by more than twice, thereby enhancing the performance of the enzymes. Li et al. [25] conducted enzymatic hydrolysis of corn starch using 30 min of ultrasound pretreatment, 50 min of liquefaction, and 90 min of saccharification. The use of ultrasound enhanced the starch hydrolysis rate, and HD values up to 70% were found in the samples submitted to ultrasound, which represents an HD 2.8 times higher than the CEH.

In this work, the SEH obtained the lowest HD. However, the ultrasound still shows a positive influence on the process performance. For example, ultrasound increased the HD by 27% and 59% in the CEH and SEH processes, respectively. As shown in Table 3, the lower HD was observed in SEH and USEH, with values of 11.43% and 28.10%, respectively. Higher HD values were obtained by Jin et al. [23], who used an SEH process in the enzymatic hydrolysis of sweet potato starch. In that work, the USEH showed the highest HD (59.10%) when compared with the SEH process (37.80%). Although simultaneous liquefaction and saccharification processes can result in a lower energy demand [4], the simultaneous approach, depending on the enzymes and conditions used, can show different results. Parameters such as temperature and pH have a key influence on enzyme activity. For example, under these conditions, the activity of α-amylase could be reduced, resulting in incomplete starch liquefaction and lower concentrations of substrates for saccharification with amyloglucosidase. For example, in the SEH, the SC was 54.75%, which indicates that the enzymes solubilized approximately half of the AER-starch. However, when the ultrasound was applied, the AER-starch was almost solubilized (91.25%). It confirms the positive effect of the ultrasound pretreatment on the enzymatic hydrolysis process.

The combination of higher SC values (>90%) and lower HD values, observed mainly in the simultaneous process, can be explained by the formation and accumulation of soluble oligosaccharides rather than by complete saccharification to glucose. SC reflects the solubilization of AER-starch into soluble compounds, whereas HD (determined by the DNS method) quantifies reducing sugars such as glucose and short-chain saccharides. In CEH and UCEH, higher SC and SC values were obtained, indicating that starch solubilization followed by saccharification was an effective process. In contrast, lower values of SC and HD were obtained in SEH because the hydrolysis was performed at a temperature outside the optimal range recommended by the enzyme supplier. Although ultrasound pretreatment enhances starch solubilization, it did not significantly promote the enzymatic conversion of soluble compounds into glucose under simultaneous hydrolysis conditions. Thus, while disruption of starch granules and generation of soluble intermediate compounds are effective during pretreatment and liquefaction, saccharification completeness is governed by temperature and pH conditions rather than starch accessibility alone.

Since enzymes were not directly exposed to ultrasound, potential cavitation-induced inactivation effects were avoided. However, the thermostability and acid tolerance of the enzymes, as well as the starch source, should be verified to guarantee good performance and enhance the overall saccharification, while maintaining the temperature and pH conditions recommended by the enzyme manufacturer. In this context, we will evaluate the effect of enzymatic hydrolysis time only in the CEH process in the following stage.

## Effect of hydrolysis time on CEH and UCEH

Process time is a key factor that influences energy consumption, and the longer the process time, the higher the energy consumption and economic cost. A preliminary set of experiments was conducted (data available on Zenodo, DOI: https://doi.org/10.5281/zenodo.18257051) to evaluate the effect of the starch concentration (10–50 g L$^{-1}$) on HD and SC. However, no significant differences were observed, suggesting that under these process conditions, the enzyme activity is enough to hydrolyze a higher amount of starch. In this context, we chosen an AER-starch concentration of 50 g L$^{-1}$ to evaluate the effect of the enzymatic hydrolysis time on the SC and HD. The following experimental conditions were evaluated for both CEH and UCEH processes: liquefaction times of 30 and 90, and 120 min; and saccharification times of 30, 90, and 120 min.

As shown in Table 4, HD values ranging from 39.48% to 76.23% were obtained. As expected, longer hydrolysis times produced higher HD values. Longer saccharification times improved the HD values. For example, for CE and UCE, when the liquefaction time was 30, and the saccharification time increased from 30 to 90 min, the HD increased from 39.48±2.55% to 64.23±2.24% and from 43.45±1.28% to 69.60±1.98% for CEH and UCEH, respectively. Longer saccharification times enhanced process performance, increasing the HD value up to 62%. A similar behavior was observed for the liquefaction time of 60 min when the saccharification time was longer in CE and UCE. In saccharification, amyloglucosidase produces glucose from oligosaccharides, and when short liquefaction times are used, initial depolymerization is limited, and intermediate products (e.g., maltose and maltodextrins) remain in the mixture. Therefore, longer saccharification time is required to complete the starch breakdown. For example, when 90 or 120 min were used in each hydrolysis stage, the higher HD values were obtained (around 75%). A similar behavior was observed during the enzymatic hydrolysis of potato starch using glucoamylase, with ultrasound pretreatment times ranging from 5 to 45 min [26]. In that research, the starch hydrolysis enhanced HD up to 28 times when the saccharification time increased from 5 to 45 min. Although according to Tukey´s test the UCE and CE are statistically similar (p<0.05), when the UCE process was carried out using a liquefaction time of 30 min and 90 min of saccharification time, the ultrasound enhanced the HD, reaching a HD of 69.60±1.98%, which is slightly higher than the same condition in CE. Although ultrasound pretreatment slightly improved the HD at reduced liquefaction time, overall, the ultrasound did not show a positive effect on HD in any of the hydrolysis times used. The amount of AER-starch used, combined with the pretreatment time, was insufficient to enhance the solubilization of the polymers through cavitation and structural breakdown. At higher starch concentrations, the solution viscosity increased, diminishing the performance of the ultrasound pretreatment. The increase in the starch solution viscosity limits the bubble growth and collapse, leading to a significant rise in the cavitation threshold [27]. Although higher

**Table 4. Summary of SC and HD from CEH and UCEH.**

|  | Liquefaction time (min) | Saccharification time (min) | HD (%) | SC (%) |
|---|---|---|---|---|
| CEH | 30 | 30 | 39.48±2.55[D] | 95.64±0.32[A] |
|  | 30 | 90 | 64.23±2.24[B] | 93.44±2.71[A] |
|  | 90 | 30 | 50.97±0.00[C] | 94.47±1.67[A] |
|  | 90 | 90 | 76.23±0.81[A] | 94.51±0.00[A] |
|  | 120 | 120 | 75.21±1.39[A] | 94.76±0.02[A] |
| UCEH | 30 | 30 | 43.45±1.28[D] | 95.93±0.08[A] |
|  | 30 | 90 | 69.60±1.98[AB] | 95.15±0.64[A] |
|  | 90 | 30 | 43.65±3.24[D] | 95.12±1.15[A] |
|  | 90 | 90 | 69.61±0.53[AB] | 94.39±0.31[A] |
|  | 120 | 120 | 67.83±0.34[B] | 95.09±0.29[A] |

Means within columns followed by different letters are significantly different (p<0.05).

amplitudes can partially compensate for the effect of high viscosity, amplitudes above 60% could not be used in this study to preserve probe integrity and ensure equipment maintenance. Consequently, ultrasound pretreatment becomes less effective at higher AER-starch concentrations.

Regarding SC, among the tested conditions, no statistical differences ($p < 0.05$) were observed. The enzymes hydrolyzed almost all the AER-starch, with SC values higher than 90% under all process conditions. However, saccharification was not complete, as indicated by the HD values observed. While SC reflects the conversion of AER-starch into soluble products, HD explains the extent to which these products have been broken down into simple sugars, detected by DNS. Therefore, these results suggest that although AER-starch was almost completely degraded under these conditions, total glucose saccharification was not achieved. In general, shorter hydrolysis times resulted in higher SC values but lower HD values, suggesting the accumulation of partially hydrolyzed oligosaccharides. In this study, although longer saccharification times increased HD values, complete saccharification to glucose was not achieved, and the positive effects of the ultrasound pretreatment were limited. Under the conditions studied, starch accessibility was unlikely to have been the limiting factor, and the saccharification step determined hydrolysis completeness. Overall, ultrasound at higher AER-starch concentrations had a limited impact on HD and no effect on SC. Therefore, the ultrasound pretreatment does not improve the hydrolysis at high AER-starch concentrations.

## Techno-economic evaluation

When the enzymatic hydrolysis process is scaled up from the laboratory to the industrial level, the AER-starch concentration has a significant impact on the manufacturing cost. Higher AER-starch concentrations can improve the process feasibility and reduce the COM. Moreover, as mentioned earlier, if the HD and SC are maintained from laboratory to industrial scale, the amount of glucose syrup produced can be increased. In this context, a techno-economic evaluation was performed using AER-starch concentrations ranging from 10 to 350 g L$^{-1}$. The higher concentration was established, according to information from the enzyme supplier.

On the other hand, ultrasound pretreatment did not significantly enhance process performance at higher AER-starch concentrations. Moreover, the higher capital cost of the UAE unit and the additional pretreatment time increase the overall process cost. For example, a preliminary analysis indicated that the COM of the UCEH process was US\$3.61 kg$^{-1}$, which is 24% higher than the COM obtained in the CEH process (US\$2.90 kg$^{-1}$) when the higher AER-starch concentration was considered. When the ultrasound pretreatment is applied at low AER-starch concentrations, process performance is enhanced by the cavitation effects that improve enzyme accessibility. However, at higher starch concentrations, the increased viscosity attenuates cavitation, reducing the propagation of ultrasonic energy. As a result, ultrasound pretreatment did not yield significant improvements in yield. Moreover, the COM increased due to higher capital and operating costs of the UAE unit, as well as additional energy consumption and pretreatment time. Therefore, under these conditions, ultrasound pretreatment is not economically feasible at industrially high AER- starch concentrations. In this context, the techno-economic assessment only considered the CE process for simulation.

The influence of the AER-starch concentrations on COM is showed in Fig 3. As expected, the higher the AER-starch concentration, the lower the COM. For example, when the AER-starch concentration increased from 10 to 350 g L$^{-1}$, the COM decreased from US\$ 62.25 kg$^{-1}$ to US\$ 2.90 kg$^{-1}$, which represents a 21 times lower COM and an increase in the volumetric productivity from 151 to 3210 kg batch$^{-1}$. Therefore, the AER-starch concentration is a key factor in the enzymatic hydrolysis processes. A deeper analysis of the COM reveals that the CRM was the primary component, making the highest contribution to COM across all AER-starch concentrations evaluated. However, its contribution was diluted at lower AER-starch concentrations as can be observed in Fig 3. This behavior can be explained by the fact that as the AER-starch concentration increases, a greater quantity of raw materials is needed to perform the process. When lower AER-starch concentrations were used, the contribution of the CRM ranged between 33.63% and 45.53%, for AER-starch concentrations of 10 and 50 g L$^{-1}$, respectively. However, the CRM contribution increased by over 50% of the total COM when

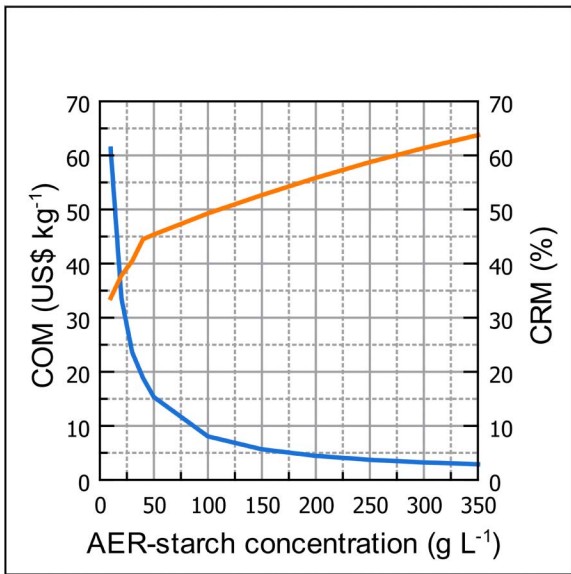

**Fig 3. Influence of the AER-starch concentration on the COM and CMR contributions to the COM.**

AER-starch concentration is above 150 g L$^{-1}$, reaching a maximum contribution of 63.75% of the COM. At the highest AER-starch concentration, for every dollar spent to produce one kilogram of glucose syrup, the raw materials account for approximately 0.63 cents. Moreover, at this AER-starch concentration, the AER-starch accounts for 67.08% of the CRM (Fig 4), followed by Sodium hydroxide (19.26%) and the buffer salts (10.13%). The enzymes are responsible for around 3% of the CRM. Therefore, at higher starch concentrations, the cost of the starch becomes a key factor in the COM. Other researchers have found that CRM is the component that contributes the most to the COM. For example, Wu et al. [28] found that the CRM represents around 87.5% of the total COM in xylose production using formic acid hydrolysis. Moreover, in the production of poly(lactic acid) (PLA) through enzymatic hydrolysis of PLA waste, the CRM is the component that makes the higher contribution to total COM, due to the value of the feedstock and the enzymes [29].

Unlike higher AER-starch concentrations, when lower starch concentrations are used, the contribution of the AER-starch to the CRW decreased to 5.73% when starch concentration is 10 g L$^{-1}$, and the use of Sodium hydroxide and buffer salts increases their contribution in CRM up to 87%, as can be observed in Fig 4. After CRM, FCI is the second component that contributes to the total COM. Contrary to the CRM, the higher the starch concentration, the lower the FCI. The contribution of the FCI to the total COM is 16.63% and 27.40% for starch concentrations of 10 and 350 g L$^{-1}$, respectively. In this process, as higher starch concentrations demand more quantities of materials, the contribution of the equipment is diluted. As shown in Fig 1, the enzymatic hydrolysis process is composed of a large quantity of equipment that requires a high initial capital investment, and this equipment incurs fixed costs based on the volumetric capacity of the process. This equipment has an associated depreciation cost that is fixed for all starch concentrations and represents 50% of the FCI.

## Sensitivity study

As stated before, the cost of the AER-starch makes a significant contribution to the COM and, therefore, to the economic feasibility of the process. Moreover, the possible selling price of the glucose syrup depends on the COM. For example, the higher the COM, the higher the selling price should be to guarantee the process feasibility. Thus, various scenarios with different AER-starch purchasing costs and glucose syrup selling prices were evaluated. Starch purchasing cost from US$1.5 kg$^{-1}$ to US$1.0 kg$^{-1}$, and selling price between US$0.5 kg$^{-1}$ to US$ 4.0 kg$^{-1}$ were considered. The gross margin is

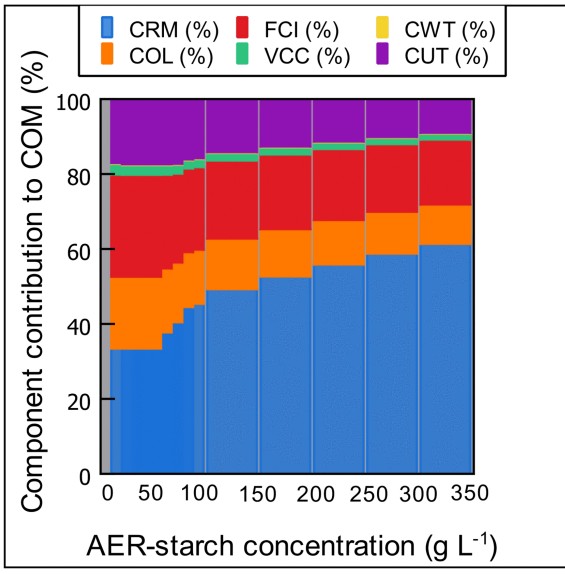

**Fig 4. Contribution of the major cost factors (CRM, COL, FCI, VCC, CWT, and CUT) to the COM.**

an economic parameter related to the percentage of the annual revenue that is gross profit [30]. The gross margin should be a positive value, and the higher the gross margin, the more attractive the project. As can be observed in Fig 5A, the gross margin becomes positive only when the glucose syrup selling price exceeds US$0.5 kg⁻¹. Moreover, when less expensive raw materials are used, the process becomes more feasible. If the AER-starch purchasing cost decreases from US$1.50 kg⁻¹ to US$1.00 kg⁻¹, the gross margin increases from 3.47% to 17.23%, representing almost five times more gross profit. Thus, the cheaper the AER starch, the more feasible the process is. Another economic parameter used in profitability evaluation is the IRR, which is similar to ROI and gross margin; the higher the IRR value, the more feasible the project. In this study, positive IRR values are found for glucose syrup selling prices higher than US$3 kg⁻¹ for all AER-starch purchasing cost. The IRR is closely related to the compound annual growth rate. For example, as shown in Fig 5B, when the AER-starch purchasing cost is US$1.0 kg⁻¹ and the glucose syrup selling price is US$4.0 kg⁻¹, the project will generate an annual return of 34.85% on the invested capital. Moreover, regardless of the AER-starch purchasing cost, according to the IRR, the process is economically feasible if the glucose syrup price is less than the US price, which means the software-calculated IRR is feasible.

The payback time is the amount of time required to recover the total capital investment. Thus, the shorter the payback time, the more attractive the project is. For all AER-starch purchasing costs, payback times can be calculated when the syrup glucose selling price exceeds US$2.50 kg⁻¹. Although the recommended payback time varies by industry, payback times of around four years are economically feasible. Lower glucose syrup does not allow calculating the payback time. Even more, as presented in Fig 5C, extremely long payback times, such as 30.55 and 16.21 years, are observed when the AER-starch purchasing cost is US$2.50 kg-1 and the selling price is also US$2.50 kg⁻¹. As expected, the shortest payback times are obtained at the lowest AER-starch purchasing cost and the highest glucose syrup selling cost. Under these conditions, the initial capital investment is recovered in 2.07 years. The ROI is an economic parameter used to evaluate the viability of a project or to compare the profitability of different projects. If the ROI has a negative value, the project is not feasible. On the contrary, positive values of ROI indicate profitability, and the higher the ROI, the more feasible the process [31]. As can be observed in Fig 5D, the ROI is positive when the glucose syrup selling price exceeds US$2.5 kg-1. However, when the glucose syrup selling cost increases and the AER-starch

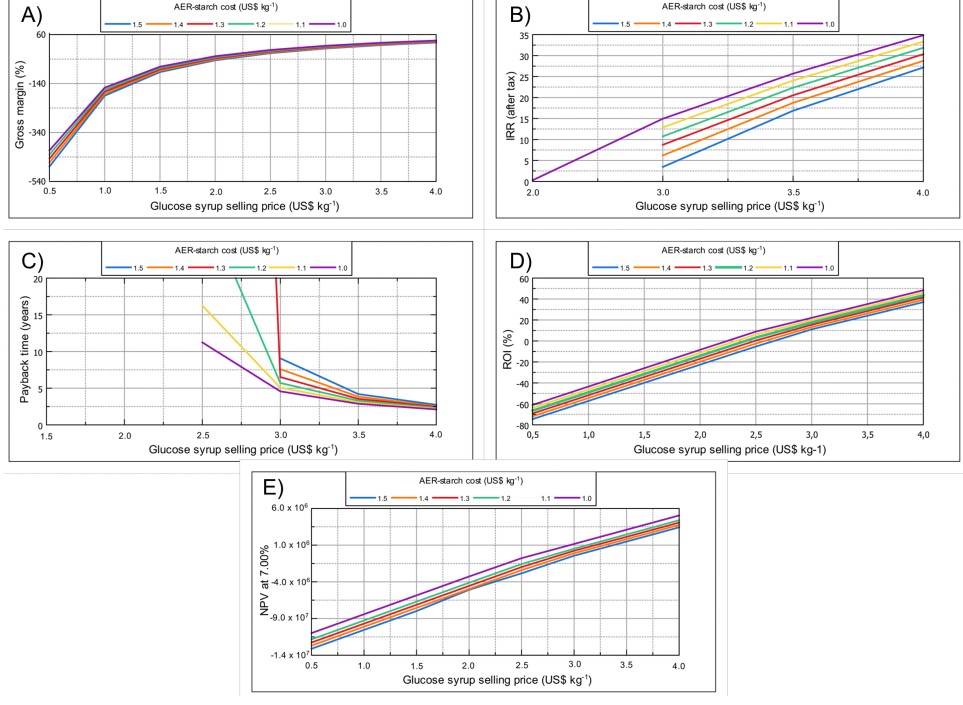

**Fig 5. Effect of the selling price of glucose syrup on performance indicators.**

purchasing cost decreases, the ROI becomes more positive. Finally, the value of future revenue cash flow compared to the initial investment is determined by the NPV. An economically feasible project should have a positive NPV. However, if the project has associated negative NPV values, it should be rejected. In this study, as shown in Fig 5D, NPV-positive values are obtained when the glucose syrup selling cost exceeds US$3.5 kg$^{-1}$, and when the AER-starch purchasing cost is between US$1.5 kg$^{-1}$ and US$1.4 kg$^{-1}$. As the CRM decreases, the process becomes more feasible, and it becomes possible to obtain positive NPV values at a lower glucose syrup selling cost. For example, when the AER-starch purchasing cost decreases to US$1.3 kg$^{-1}$, the NPV becomes positive (US$231,035) when the glucose syrup selling cost of US$3.0 kg$^{-1}$ is used. Moreover, the highest NPV value is obtained when the highest syrup selling cost and the lowest AER-starch purchasing cost are used. Under these conditions, as can be observed in Fig 5, the project would generate a net gain of US$5,041,458 after the discount rate (7%).

In summary, glucose syrup production is economically feasible when less expensive raw materials are available, especially when the AER-starch purchasing cost is US$0.50 kg$^{-1}$, resulting in strong profitability even at glucose syrup selling prices of US$2.00 kg$^{-1}$. However, when the AER-starch cost increases above US$0.50 kg$^{-1}$, the process feasibility decreases, especially when the AER-starch cost reaches US$1.50 kg$^{-1}$. The project is feasible if the glucose syrup is sold for US$4.00 kg$^{-1}$ or more. Therefore, in the scenarios evaluated, the process can only be feasible if the AER-starch purchasing cost is at least US$1.20 kg$^{-1}$ and the glucose syrup selling price of a minimum of US$3.50 kg$^{-1}$ is competitive in the market. Moreover, commercial glucose syrup is commonly produced from corn or cassava starch via enzymatic hydrolysis with α-amylase and glucoamylase, using a well-established industrial route. While the conventional process benefits from economies of scale, only conventional refined starch feedstocks are used [32]. The production of glucose syrup from AER-starch could represent an alternative pathway for valorizing extraction residues and intensify the use of raw materials, contributing to the circular economy. Although the proposed enzymatic hydrolysis process is similar to the conventional process, using AER-starch may offer greater sustainability and reduced waste.

Nevertheless, in a real scenario, factors such as AER-starch composition and equipment downtime can pose risks to the process implementation. For example, AER-starch composition (e.g., moisture content, impurities, etc) can affect the efficiency of the hydrolysis. Therefore, at the industrial level, appropriate quality control of raw materials should be established. Moreover, equipment downtime associated with ultrasound pretreatment, the enzymatic reactor vessel, and evaporation could affect plant productivity. Therefore, factors such as preventive maintenance and process monitoring should be considered for process implementation in a real scenario.

## Conclusions

Ultrasound pretreatment enhances the enzymatic hydrolysis at AER-starch concentrations ranging from 10 to 30 g L$^{-1}$, although its effectiveness diminishes at higher concentrations. HD and SC varied from 11.43% to 76.23% and from 54.75% to 96.77%, respectively. Almost all AER-starch was solubilized during the enzymatic hydrolysis, and the CEH showed the highest starch conversion in soluble sugars when the liquefaction and saccharification times were 90 and 120 min. The lowest HD values were observed when liquefaction and saccharification were performed simultaneously, resulting in a 66% reduction in sugar production. The techno-economic evaluation revealed that when the AER-starch concentration increases, the COM decreases. The COMs for the glucose syrup were US$61.25 kg$^{-1}$ and US$2.90 kg$^{-1}$ at AER-starch concentrations of 10 and 350 g L$^{-1}$, respectively. At higher AER-starch concentrations, the main component of the COM was the CRM, which represents above 50% in AER-starch concentrations higher than 100 g L$^{-1}$. AER-starch constituted the main cost in raw materials, being responsible for up to 67.08% of the CRM when an AER-starch concentration of 350 g L$^{-1}$ is used. According to the sensitivity study, higher ROI, gross margin, NPV, and IRR are obtained when the selling price of the glucose syrup increases and the purchase price of the AER-starch decreases. Unlike ROI, gross margin, NPV, and IRR, shorter payback times are obtained under these conditions. The enzymatic hydrolysis of AER-starch was economically feasible, with an AER-starch purchasing cost of US$1.50 kg$^{-1}$, provided the selling price of the glucose syrup exceeded US$4.00 kg$^{-1}$. However, the availability of less expensive raw materials allows for establishing lower selling prices (US$3.50 kg$^{-1}$) when the purchasing of the AER-starch is lower than US$1.20 kg$^{-1}$.

## Supporting information

**S1 Table. Economic assumptions for the enzymatic hydrolysis process.** This Table summarizes the parameters used in the techno-economic analysis.
(DOCX)

**S2 Table. Summarized mass and energy balances for the hydrolysis process.** This Table summarizes the major material flows and utility demands obtained from the SuperPro Designer v14® simulation.
(DOCX)

## Author contributions

**Conceptualization:** Susana Ochoa, Víctor-Manuel Osorio-Echeverri, J. Felipe Osorio-Tobón.

**Formal analysis:** J. Felipe Osorio-Tobón.

**Funding acquisition:** J. Felipe Osorio-Tobón.

**Investigation:** J.D. Quiñonez-Ensuncho.

**Methodology:** J.D. Quiñonez-Ensuncho, J. Felipe Osorio-Tobón.

**Writing – original draft:** J. Felipe Osorio-Tobón.

**Writing – review & editing:** J. Felipe Osorio-Tobón.

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
