## [Decision Letter · Decision Letter 0]

9 Dec 2025

Dear Dr. Osorio-Tobón,

Thank you for submitting your manuscript to PLOS ONE. After careful consideration, we feel that it has merit but does not fully meet PLOS ONE’s publication criteria as it currently stands. Therefore, we invite you to submit a revised version of the manuscript that addresses the points raised during the review process.

We look forward to receiving your revised manuscript.

Kind regards,

Leonidas Matsakas

Academic Editor

PLOS One

Journal Requirements:

“Institución Universitaria Colegio Mayor de Antioquía. Convocatoria de investigación. FCSA23.”

“The authors are grateful to COLMAYOR for financial support.”

“Institución Universitaria Colegio Mayor de Antioquía. Convocatoria de investigación. FCSA23.”

5. We note that your Data Availability Statement is currently as follows: All relevant data are within the manuscript and its Supporting Information files.

Reviewers' comments:

Reviewer's Responses to Questions

**Comments to the Author**

1. Is the manuscript technically sound, and do the data support the conclusions?

Reviewer #1: Yes

Reviewer #2: Yes

2. Has the statistical analysis been performed appropriately and rigorously?

Reviewer #1: Yes

Reviewer #2: I Don't Know

3. Have the authors made all data underlying the findings in their manuscript fully available?

Reviewer #1: Yes

Reviewer #2: No

4. Is the manuscript presented in an intelligible fashion and written in standard English?

Reviewer #1: Yes

Reviewer #2: Yes

Reviewer #1: The manuscript explores ultrasound pretreatment to enhance enzymatic hydrolysis of starch recovered from purple yam extraction residues. consecutive enzymatic hydrolysis (CEH) and simultaneous enzymatic hydrolysis (SEH) were tested with and without ultrasound, and hydrolysis degree, starch conversion, and techno economic feasibility were evaluated using SuperPro Designer. The authors report that ultrasound improves hydrolysis at low starch concentrations, CEH outperforms SEH, and economic viability depends on feedstock cost and glucose syrup price. Although the work is relevant and integrates experimental and economic analysis, the manuscript needs major revision to improve clarity, methodological detail, interpretation consistency, and transparency of the economic assessment.

Review comments:

• The manuscript would benefit from a clearer mechanistic explanation of the ultrasound pretreatment, particularly at higher starch concentrations, to strengthen the scientific rationale presented in the Introduction section.

• A more detailed discussion of the observed combination of high SC and lower HD would enrich the understanding of potential oligosaccharide formation and saccharification completeness.

• Sections 2.2 & 2.3, Including thermal-only controls and confirming enzyme stability under SEH conditions would enhance the robustness of the experimental design and improve interpretation of ultrasound-specific effects.

• Section 2.3, Reporting additional ultrasound parameters such as delivered power, acoustic energy density, and temperature control would support reproducibility and alignment with literature.

• Section 3.1, Providing a more detailed explanation for cases where SC exceeds 90% but HD remains lower would enrich understanding of oligosaccharide formation and saccharification completeness.

• Sections 3.1 & 3.2, Clarifying the varying influence of ultrasound across experiments would help present a more cohesive interpretation, as SEH and CEH results differ in several places.

• Including mass and energy balance summaries would support validation of the process model and clarify how each unit operation contributes to overall process performance.

• In Techno Economic Evaluation (Section 2.6), Providing justification for key assumptions such as labor cost, resin replacement rates, evaporation energy, and depreciation, would improve transparency of the economic modeling.

• In the TEA Results and Sensitivity Analysis, supporting assumed AER starch prices with market references would make the economic conclusions more robust and credible.

• Discussing scale-up factors such as mixing, heat transfer, enzyme kinetics, and viscosity would improve the realism of extrapolating laboratory performance to industrial conditions.

• Expanding the literature review to include recent advances in ultrasound-assisted enzymatic hydrolysis and cavitation mechanisms would enhance the scientific context in the Introduction section.

• Including a brief comparison with existing commercial glucose syrup production routes could further strengthen the TEA by highlighting the competitive position and market potential of glucose syrup derived from AER starch.

• Incorporating a short discussion on operational risks, such as feedstock variability or potential equipment downtime, would add practical context and help readers appreciate real-world implementation factors.

Reviewer #2: The manuscript is well organized and documents all experimental protocols. However, the process modelling and technoeconomic assessment (TEA) lack standard documentation of modelling and economic assumptions, which makes it difficult to follow the TEA results.

I suggest the authors consider revising the following aspects:

Section 2.5

Expand/clarify process modeling details. A paragraph can be added to describe the following:

- Thermodynamic/physical property methods used in SuperPro Designer

- How non-conventional components such as AER-starch and protiens were represented. Pseudo-components or user defined?

- Whether full energy balances were solved, how electricity/steam demands were estimated for the evaporation, filtration and mixing steps

Section 3.3

Strengthen the justification for excluding ultrasound in the TEA flowsheet. It was mentioned that ultrasound at high starch concentration does not significantly improve performance and would increase COM; however, in the results it was stated only that ultrasound was not considered further.

You could:

- Include a simplified alternative flowsheet with ultrasound units (even if only described qualitatively).

- Provide a short comparative COM estimate (e.g., “preliminary analysis indicated that including ultrasound increased COM by X–Y% at 350 g·L⁻¹ with no significant yield benefit”).

- Clearly state why ultrasound is attractive at low concentrations (process intensification) but not at the industrially relevant high concentrations.

Section 3.4

Clarify the economic framework more systematically. Right now, elements like discount rate (7%), payback, ROI, IRR, and NPV are described in the results, but not all main cash-flow assumptions are grouped in one place.

I’d suggest to add a small “Economic assumptions” table/box containing:

- Project lifetime (years).

- Discount rate.

- Tax rate (or explicitly state that taxes are neglected).

- Depreciation method and period.

- Working capital assumptions (if any).

- Currency and reference year for all costs (e.g., “all costs in 2024 USD”).

- Basis (330 days/year, 5000 L batch, plant availability, etc.).

That makes the TEA more standard and easier to compare with other studies.

**Do you want your identity to be public for this peer review?** For information about this choice, including consent withdrawal, please see our Privacy Policy

Reviewer #1: No

Reviewer #2: **Yes:** Sennai Mesfun

---

## [Author Response · Author response to Decision Letter 1]

15 Jan 2026

Journal Requirements: The reply and manuscript’s text adjustments of Reviewer #1 comments were highlighted in green. We thank the academic editor and PLOS ONE staff thorough review of the manuscript.

Reply: Thank you for your feedback. We have reviewed the style requirements.

Reply: Thank you for your feedback. All relevant data are provided within the manuscript. The process simulations were performed using SuperPro Designer v14®, a commercially licensed software. No author-generated code was used in this study. As the software is proprietary, it cannot be shared. However, all model parameters, assumptions, and input data required to reproduce the simulations have been fully described and are available in the manuscript.

“Institución Universitaria Colegio Mayor de Antioquía. Convocatoria de investigación. FCSA23.”

Reply: Thank you for your feedback. We have reviewed it.

“The authors are grateful to COLMAYOR for financial support.”

“Institución Universitaria Colegio Mayor de Antioquía. Convocatoria de investigación. FCSA23.”

Reply: Thank you for your feedback. We have reviewed it.

5. We note that your Data Availability Statement is currently as follows: All relevant data are within the manuscript and its Supporting Information files.

Reply: Thank you for your feedback. We confirm that all raw data required to replicate the results of our study have been made publicly available. The complete dataset is openly accessible on Zenodo at the following DOI:

https://doi.org/10.5281/zenodo.18257051

Reply: Thank you for your comment. We have revised the manuscript accordingly. The phrase “data not shown” has been removed and all data have now been made publicly available. These data are included in the dataset deposited in Zenodo, accessible at the following DOI:

https://doi.org/10.5281/zenodo.17872033

Reply: Thank you for your feedback. We have reviewed it.

Reviewers' comments:

Reviewer #1: The reply and manuscript’s text adjustments of Reviewer #1 comments were highlighted in yellow. We thank the Reviewer for the thorough review of the manuscript.

1. The manuscript would benefit from a clearer mechanistic explanation of the ultrasound pretreatment, particularly at higher starch concentrations, to strengthen the scientific rationale presented in the Introduction section.

Reply: Thank you for your feedback. We have added the following paragraph to the discussion:

At higher starch concentrations, the solution viscosity increased, diminishing the performance of the ultrasound pretreatment. The increase in the starch solution viscosity limits the bubble growth and collapse, leading to a significant rise in the cavitation threshold [27]. Although higher amplitudes can partially compensate for the effect of high viscosity, amplitudes above 60% could not be used in this study to preserve probe integrity and ensure equipment maintenance. Consequently, ultrasound pretreatment becomes less effective at higher AER-starch concentrations.

2. A more detailed discussion of the observed combination of high SC and lower HD would enrich the understanding of potential oligosaccharide formation and saccharification completeness.

Reply: Thank you for your feedback. We have added the following paragraph to the discussion:

In general, shorter hydrolysis times resulted in higher SC values but lower HD values, suggesting the accumulation of partially hydrolyzed oligosaccharides. In this study, although longer saccharification times increased HD values, complete saccharification to glucose was not achieved, and the positive effects of the ultrasound pretreatment were limited. Under the conditions studied, starch accessibility was unlikely to have been the limiting factor, and the saccharification step determined hydrolysis completeness. Overall, ultrasound at higher AER-starch concentrations had a limited impact on HD and no effect on SC. Therefore, the ultrasound pretreatment does not improve the hydrolysis at high AER-starch concentrations.

3. Sections 2.2 & 2.3, Including thermal-only controls and confirming enzyme stability under SEH conditions would enhance the robustness of the experimental design and improve interpretation of ultrasound-specific effects.

Reply: Thank you for this observation. The SEH was explored as a strategy to reduce overall processing time by combining the liquefaction and saccharification steps. However, the hydrolysis yields obtained under SEH conditions were lower than those achieved in the sequential process. This behavior can be explained by the different optimal temperature requirements of the enzymes employed. The α-amylase used is an industrial enzyme designed to operate at high temperatures (90 °C), whereas amyloglucosidase exhibits optimal activity at lower temperatures (60 °C). Operating SEH at an intermediate temperature (75 °C) represented a compromise that likely reduced amyloglucosidase activity and limited saccharification efficiency as we mentioned in the text. These results highlight the importance of enzyme thermal compatibility when implementing SEH, rather than indicating intrinsic limitations of the process or enzyme performance.

4. Section 2.3, Reporting additional ultrasound parameters such as delivered power, acoustic energy density, and temperature control would support reproducibility and alignment with literature.

Reply: Thank you for your feedback. We have added the following information to the manuscript:

Ultrasound pretreatment was performed using a 750 W ultrasonic homogenizer (Cole-Parmer, Vernon Hills, IL, USA) equipped with temperature control (30°C). AER-starch mixtures were treated for 30 minutes at an amplitude of 60% in pulse mode (2 seconds ON, 2 seconds OFF), with an estimated delivered power of 600 W and an acoustic energy density of 592 W cm-².

5. Section 3.1, Providing a more detailed explanation for cases where SC exceeds 90% but HD remains lower would enrich understanding of oligosaccharide formation and saccharification completeness.

Reply: Thank you for your feedback. We have added the following information to the manuscript:

The combination of higher SC values (>90%) and lower HD values, observed mainly in the simultaneous process, can be explained by the formation and accumulation of soluble oligosaccharides rather than by complete saccharification to glucose. SC reflects the solubilization of AER-starch into soluble compounds, whereas HD (determined by the DNS method) quantifies reducing sugars such as glucose and short-chain saccharides. In CEH and UCEH, higher SC and SC values were obtained, indicating that starch solubilization followed by saccharification was an effective process. In contrast, lower values of SC and HD were obtained in SEH because the hydrolysis was performed at a temperature outside the optimal range recommended by the enzyme supplier. Although ultrasound pretreatment enhances starch solubilization, it did not significantly promote the enzymatic conversion of soluble compounds into glucose under simultaneous hydrolysis conditions. Thus, while disruption of starch granules and generation of soluble intermediate compounds are effective during pretreatment and liquefaction, saccharification completeness is governed by temperature and pH conditions rather than starch accessibility alone.

6. Sections 3.1 & 3.2, Clarifying the varying influence of ultrasound across experiments would help present a more cohesive interpretation, as SEH and CEH results differ in several places.

Reply: Thank you for your feedback. We agree that clarifying the differing influence of ultrasound across CEH and SEH is important for a cohesive interpretation of the results. Based on your suggestion and related comments from the other reviewer, we revised Sections 3.1 and 3.2 to explicitly distinguish the role of ultrasound under each hydrolysis strategy.

7. Including mass and energy balance summaries would support validation of the process model and clarify how each unit operation contributes to overall process performance.

Reply: Thank you for the comment. This information has been compiled and is now presented as supplementary material in S2 Table.

8. In Techno Economic Evaluation (Section 2.6), Providing justification for key assumptions such as labor cost, resin replacement rates, evaporation energy, and depreciation, would improve transparency of the economic modeling.

Reply: Thank you for the comment. This information has been compiled and is now presented as supplementary material in S1 Table.

9. In the TEA Results and Sensitivity Analysis, supporting assumed AER starch prices with market references would make the economic conclusions more robust and credible.

Reply: Thank you for your feedback. In the market, AER-starch is not commercially available on the market. Thus, its purchasing cost was estimated based on the price of cassava starch, which can be considered a comparable product. In this context, the price was obtained from a local grocery store. We have added the market reference to the manuscript.

10. Discussing scale-up factors such as mixing, heat transfer, enzyme kinetics, and viscosity would improve the realism of extrapolating laboratory performance to industrial conditions.

Reply: Thank you for your feedback. We have added the following information to the manuscript:

Factors such as mixing and heat transfer, as well as viscosity and enzyme kinetics, should be considered during scale-up. For example, an increase in viscosity can reduce bubble collapse, reducing the cavitation effect and mass transfer rates. Moreover, the distribution of ultrasound energy is influenced by reactor size because cavitation is not uniform across the reactor. Moreover, an increase in temperature requires precise temperature control to preserve enzyme activity [18]. On the other hand, as acoustic cavitation may negatively affect enzyme stability, the ultrasound is commonly applied as a pretreatment rather than simultaneously.

In practice, perfect mixing conditions cannot be achieved at a large scale, and there is no single consensus approach for scale-up [19]. Instead, a scale-up approach could rely on maintaining comparable laboratory-scale operating conditions and assuming they can be extrapolated, while acknowledging unavoidable gradients in temperature, pH, and concentration at the industrial scale. In this approach, we assumed that the HD and SC of the hydrolysates obtained at the laboratory scale would be reproduced at larger scales under the same hydrolysis conditions (e.g., AER-starch concentration, hydrolysis time, temperature, etc.).

11. Expanding the literature review to include recent advances in ultrasound-assisted enzymatic hydrolysis and cavitation mechanisms would enhance the scientific context in the Introduction section.

Reply: Thank you for your feedback. We have added the following information to the manuscript:

Depending on the matrix and enzyme, the ultrasound operation parameters have different effects. For example, during the preparation of hydrolyzed egg yolk powder after ultrasound pretreatment, cavitation and excessive ultrasound led to gradual increases and decreases in surface hydrophobicity and free sulfhydryl groups, as observed with increasing ultrasound time. In consequence, the protein swollen with disulfide bonds becomes more aggregated, and the reconstruction of these bonds was facilitated by cavitation [11]. Cavitation also generates microjets and radicals, thereby increasing selective disruption of the cell wall in lignoc

---

## [Decision Letter · Decision Letter 1]

4 Feb 2026

Dear Dr. Osorio-Tobón,

Thank you for submitting your manuscript to PLOS ONE. After careful consideration, we feel that it has merit but does not fully meet PLOS ONE’s publication criteria as it currently stands. Therefore, we invite you to submit a revised version of the manuscript that addresses the points raised during the review process.

We look forward to receiving your revised manuscript.

Kind regards,

Leonidas Matsakas

Academic Editor

PLOS One

Journal Requirements:

Reviewers' comments:

Reviewer's Responses to Questions

**Comments to the Author**

Reviewer #1: All comments have been addressed

Reviewer #2: All comments have been addressed

2. Is the manuscript technically sound, and do the data support the conclusions?

Reviewer #1: Yes

Reviewer #2: Yes

3. Has the statistical analysis been performed appropriately and rigorously?

Reviewer #1: Yes

Reviewer #2: N/A

4. Have the authors made all data underlying the findings in their manuscript fully available?

Reviewer #1: Yes

Reviewer #2: Yes

5. Is the manuscript presented in an intelligible fashion and written in standard English?

Reviewer #1: Yes

Reviewer #2: Yes

Reviewer #1: The revision has addressed the main concerns and substantially improved clarity and TEA transparency. However, minor revisions are required before the manuscript can be accepted. The remaining points are as follows:

• Please standardize units/notation (e.g., g/L, kg/kg, US$/kg) and define all acronyms at first mention (AER, SC, HD, CEH/SEH/UCEH) to improve readability.

• Please add a brief justification (or citation) for the selected enzyme dosages and hydrolysis duration to strengthen comparability with prior studies.

• Please clarify the intended syrup product specification (e.g., target solids content and typical sugar profile) so the selling-price assumption is easier to interpret.

• For transparency, please specify whether costs are reported in constant-year USD and include the price year/inflation basis in Table S1 to support reproducibility.

Reviewer #2: (No Response)

**Do you want your identity to be public for this peer review?** For information about this choice, including consent withdrawal, please see our Privacy Policy

Reviewer #1: No

Reviewer #2: **Yes:** Sennai Mesfun

---

## [Author Response · Author response to Decision Letter 2]

9 Feb 2026

Review Comments to the Author:

Reviewer #1: The reply and manuscript’s text adjustments of Reviewer #1 comments were highlighted in yellow. We thank the Reviewer for the thorough review of the manuscript.

• Please standardize units/notation (e.g., g/L, kg/kg, US$/kg) and define all acronyms at first mention (AER, SC, HD, CEH/SEH/UCEH) to improve readability.

Reply: Thank you for the comment. We have revised the manuscript to standardize all units and notation across the text, tables, and figures. Additionally, all acronyms and abbreviations were revised.

• Please add a brief justification (or citation) for the selected enzyme dosages and hydrolysis duration to strengthen comparability with prior studies.

Reply: Thank you for the comment. The enzyme dosages and hydrolysis durations were selected based on the technical datasheets and usage recommendations provided by the enzyme supplier (Proenzimas). This information has been included in the Materials and Methods section, together with a reference to the supplier’s technical documentation.

• Please clarify the intended syrup product specification (e.g., target solids content and typical sugar profile) so the selling-price assumption is easier to interpret.

Reply: Thank you for the comment. We assumed a final composition of the glucose syrup as approximately 90% glucose, 1.2% maltose, and 1.2% maltotriose, with traces of minor soluble and proteins. This information was added to the text.

• For transparency, please specify whether costs are reported in constant-year USD and include the price year/inflation basis in Table S1 to support reproducibility.

Reply: Thank you for the comment. We have added the following information to Table S1:

Reference year: 2025 (all costs expressed in constant 2025 USD.

Inflation index: 2025 (the capital cost was adjusted using SuperPro V14 built-in values of the Chemical Engineering Cost Index).

Reviewer #2: (No Response)

---

## [Editor Report · Decision Letter 2]

15 Feb 2026

Enzymatic hydrolysis of starch from the anthocyanin extraction residue (AER-starch) with ultrasound pretreatment: a techno-economic assessment

PONE-D-25-60146R2

Dear Dr. Osorio-Tobón,

We’re pleased to inform you that your manuscript has been judged scientifically suitable for publication and will be formally accepted for publication once it meets all outstanding technical requirements.

Kind regards,

Leonidas Matsakas

Academic Editor

PLOS One
---

## [Editor Report · Acceptance letter]

PONE-D-25-60146R2

PLOS One

Dear Dr. Osorio-Tobón,

I'm pleased to inform you that your manuscript has been deemed suitable for publication in PLOS One. Congratulations! Your manuscript is now being handed over to our production team.

Kind regards,

on behalf of

Dr. Leonidas Matsakas

Academic Editor

PLOS One